# Melanoma Detection by Non-Specialists: An Untapped Potential for Triage?

**DOI:** 10.3390/diagnostics12112821

**Published:** 2022-11-16

**Authors:** Carmen Cantisani, Luca Ambrosio, Carlotta Cucchi, Fanni Adél Meznerics, Norbert Kiss, András Bánvölgyi, Federica Rega, Flavia Grignaffini, Francesco Barbuto, Fabrizio Frezza, Giovanni Pellacani

**Affiliations:** 1Dermatology Clinic, Department of Clinical Internal, Anesthesiologic and Cardiovascular Sciences, Sapienza Medical School, Sapienza University of Rome, 00185 Rome, Italy; 2Department of Dermatology, Venereology and Dermatooncology, Semmelweis University, 1085 Budapest, Hungary; 3Department of Information Engineering, Electronics and Telecommunications, Sapienza University of Rome, 00184 Rome, Italy

**Keywords:** melanoma screening, triage, artificial intelligence

## Abstract

Introduction: The incidence of melanoma increased considerably in recent decades, representing a significant public health problem. We aimed to evaluate the ability of non-specialists for the preliminary screening of skin lesions to identify melanoma-suspect lesions. Materials and Methods: A medical student and a dermatologist specialist examined the total body scans of 50 patients. Results: The agreement between the expert and the non-specialist was 87.75% (κ = 0.65) regarding the assessment of clinical significance. The four parameters of the ABCD rule were evaluated on the 129 lesions rated as clinically significant by both observers. Asymmetry was evaluated similarly in 79.9% (κ = 0.59), irregular borders in 74.4% (κ = 0.50), color in 81.4% (κ = 0.57), and diameter in 89.9% (κ = 0.77) of the cases. The concordance of the two groups was 96.9% (κ = 0.83) in the case of the detection of the Ugly Duckling Sign. Conclusions: Although the involvement of GPs is part of routine care worldwide, emphasizing the importance of educating medical students and general practitioners is crucial, as many European countries lack structured melanoma screening training programs targeting non-dermatologists.

## 1. Introduction

The incidence of melanoma increased considerably in recent decades [1]. The prognosis is directly related to the tumor’s stage; therefore, the importance of early diagnosis is undisputable [2,3].

Scientific evidence suggests that macroscopic—so-called naked eye—assessment alone achieves a diagnostic accuracy of 70% in the diagnosis of melanoma [4]. The high risk of false positives and the moderate but more dangerous risk of incorrect negative diagnoses have led to the development of instrumental techniques. The most widely used technique, dermoscopy, increases the diagnostic accuracy to 80–90% [5].

To standardize the diagnosis of melanoma, numerous clinical criteria and diagnostic techniques have been developed to help macroscopic and dermoscopic evaluation, including the ABCD rule [6] of macroscopic assessment and the three-point and seven-point checklists [7,8] for dermoscopic evaluation. Specific signs characterizing the tumor macroscopically have also been defined, such as the Ugly Duckling Sign, referring to a suspect naevus with different characteristics than the other nevi of the same patient, and the Little Red Riding Hood Sign, characterizing amelanotic melanomas that appear pink, reddish or purplish red in color [9,10,11].

Additional techniques, like total-body photography (TBP), also aim to help with diagnosis. TBP allows the acquisition of photographs of the entire body, enabling the mapping of lesions and monitoring of the patient’s entire skin surface [12]. TBP is particularly useful for patients with a history of melanoma or a family history of melanoma and those with numerous or atypical nevi [13]. It can help identify new lesions and monitor those that have changed over time [13].

The involvement of general practitioners (GPs) in melanoma screening is already part of routine care in countries where the incidence of skin tumors is particularly high [14,15,16,17]. Although several initiatives in Europe are urging GP involvement, the practice is not widespread in many countries [18,19,20,21].

The intensive development of Artificial Intelligence (AI) algorithms increased the number of available AI-based tools exponentially in the last decade. Every field, including medicine, is undergoing profound revolutions due to technological advances. Combining TBD with software using artificial intelligence, lesions can be classified based on the risk of malignancy, enabling the targeting of suspect lesions requiring excisional biopsy and histologic examination for diagnosis [22,23]. These presently evolving, artificial intelligence-based diagnostic tools might urge the inclusion of non-specialist doctors in melanoma screening, representing an untapped potential for efficient and more widely available triage.

We aimed to compare the screening efficacy of a non-specialist and a professional dermatologist.

## 2. Materials and Methods

### 2.1. Study Design and Data Retrieval

Fifty patients were randomly selected from the hospital information system (HIS) database of the outpatient department of nevi and skin tumors in the UOC of Dermatology of the Azienda Ospedaliero Universitaria Policlinico Umberto I of Rome. The patients underwent a dermatological examination and a total body scan with a VIDIX 4.0 video dermatoscope.

The scans were examined by a non-specialist (a medical student after a one-month-long training period in dermoscopic techniques without previous experience in oncological dermatology and dermoscopy) and an expert (an experienced dermatologist). Both were asked to identify clinically significant lesions on the scans. The lesions rated as clinically significant by both an expert and non-specialist were then evaluated using the ABCD rule and looking for the Ugly Duckling and Little Red Riding Hood signs. The dermoscopic images of the identified suspect lesions were examined using the three-point checklist. A total of 800 digital images and 126 dermoscopic images were analyzed.

### 2.2. Statistical Analysis

Cohen’s Kappa (κ) was calculated to assess the degree of agreement regarding the following characteristics: clinical significance of the lesion; presence of ABCD features; presence of the Ugly Duckling sign; presence of the Little Red Riding Hood sign; presence of three-point checklist signs. Specificity, sensitivity, positive predictive value, and negative predictive value were also calculated; true positive and true negative results were defined based on the expert’s assessment of the specific criteria. Statistical analyses were performed using Statistica v13.5.0.17 software (TIBCO Software Inc., Palo Alto, CA, USA).

## 3. Results

### 3.1. Macroscopic Evaluation

The results of the macroscopic evaluation are detailed in Table 1. Regarding the assessment of clinical significance, the agreement between the expert and the non-specialist was 87.75% (κ = 0.65). The four parameters of the ABCD rule were evaluated on the 129 lesions rated as clinically significant by both observers. Asymmetry was evaluated similarly in 79.9% (κ = 0.59), irregular borders in 74.4% (κ = 0.50), color in 81.4% (κ = 0.57), and diameter in 89.9% (κ = 0.77) of the cases. The concordance of the two groups was 96.9% (κ = 0.83) in the case of the detection of the Ugly Duckling Sign, while the Little Red Riding Hood sign was only detected once in the whole examined population (κ = 1.00).

The accuracy of a non-specialist for the assessment of the clinical significance of a lesion was considerably great, with a sensitivity of 83.23% and a negative predictive value of 95.66%. The assessment of the characteristics involved in the ABCD rule showed more modest results. Asymmetry and diameter evaluation had the highest sensitivity and negative predictive value, while irregular borders and color were harder to assess. The Ugly Duckling Sign was identified with the most heightened sensitivity and a negative predictive value, while the Little Red Riding Hood sign was detected only in one case (see Table 1).

### 3.2. Dermoscopic Evaluation

The dermoscopic assessment was performed using the three-point checklist; the results are detailed in Table 2. Asymmetry was assessed similarly in 84.1% (κ = 0.68), atypical network in 81.7% (κ = 0.57), while the blue veil was assessed with a concordance of 96.8% (κ = 0.78) of the cases. The sensitivity and the negative predictive value were 84.48% and 83.36% in the detection of asymmetry, 80.00% and 98.28% in the detection of the blue veil, respectively, while the atypical network was harder to assess, with a specificity of 65.85%, and a negative predictive value of 84.44%.

## 4. Discussion

The increasing incidence of melanoma highlights the importance of widely accessible screening, which is limited by the number of dermatologist specialists [1]. Our results suggest that the involvement of non-specialist physicians, such as GPs, in melanoma screening could be a potential solution in the future, representing an untapped potential for patient triage. Nevertheless, it is crucial to consider the possible downfalls as well.

The most important characteristics of a test for melanoma screening are the sensitivity and the negative predictive value [24], as false negative cases represent the most significant threat since the late diagnosis of melanoma increases mortality [2,3]. Dermoscopy is the most widely used non-invasive tool in the field of dermatology, with a sensitivity of 90% for melanoma diagnosis [25]. Although the efficacy of the new wave of imaging techniques, such as reflectance confocal microscopy (RCM), optical coherence tomography (OCT), and LED-based devices, is also heavily investigated, the implementation of these techniques in clinical practice requires effective pre-screening to enable the imaging of suspect lesions only [26,27,28].

According to our results, non-specialists are able to determine the clinical significance of a lesion with considerably great sensitivity and negative predictive value compared to the expert. While detecting irregular borders and color requires more clinical experience, asymmetry and diameter seemed easier to assess. The Ugly Duckling Sign showed the best result among the examined criteria, while the Little Red Riding Hood sign was detected only in one case; therefore, further conclusions cannot be drawn on its effectiveness in melanoma detection by non-specialists.

The dermoscopic assessment of the suspect lesions with the three-point checklist showed surprisingly excellent results considering the profound experience needed for accurate dermoscopic diagnosis in general [5]. The data show that the blue veil and asymmetry are the parameters most easily detected by the non-specialist while detecting an atypical network represents a more significant challenge.

Our results align with previous studies assessing the specificity and sensitivity of GPs for melanoma screening [14,29]. Although we found similar results when assessing the diagnostic accuracy of macroscopic and dermoscopic evaluation separately, previous studies reported that combining naked-eye and dermoscopic assessment can increase diagnostic accuracy [19,30,31,32,33]. However, as the first step of non-specialist involvement in screening, implementing macroscopic assessment of the whole skin surface of the patients in daily practice would be more reasonable than implementing dermoscopic assessment, as it has no special asset requirements [31,33]. In addition, higher sensitivity can be achieved by combining the use of the ABCD rule and the Ugly Duckling Sign detection [11]. Overall, our results highlight the untapped potential of non-specialists, such as GPs, in triage.

In light of these results, it would be worthwhile to determine whether a more in-depth training course would further improve the performance of the non-specialist. Training medical students to screen patients based on simple criteria can enhance their efficacy to triage as future doctors. Furthermore, the impact of innovative technologies should be assessed with a particular focus on artificial intelligence techniques capable of supporting the diagnosis. Further efforts are needed to identify more effective criteria to increase efficacy and reduce the number of unnecessary visits to specialists.

One of the main limitations of our study is the relatively small sample size. Only one medical student and one dermatologist specialist performed the examination; therefore, further studies are required to strengthen the quality of evidence. Additionally, it is important to note that the true positive and true negative results used for calculating specificity, sensitivity, and positive and negative predictive values are based on the expert’s judgment, not on the histological results.

## 5. Conclusions

Although the involvement of general practitioners is part of routine clinical care worldwide, emphasizing the importance of educating medical students and GPs is crucial, as many European countries lack structured melanoma screening training programs targeting non-dermatologists. 

## Figures and Tables

**Table 1 diagnostics-12-02821-t001:** Results of the macroscopic assessment by the expert and the non-specialist.

	True Positive ^a^	False Positive ^b^	True Negative ^c^	False Negative ^d^	Total	Specificity	Sensitivity	Positive Predictive Value	Negative Predictive Value	Cohen’s Cappa
Clinical significance	129	72	573	26	800	88.84%	83.23%	64.18%	95.66%	0.65
Asymmetry	46	11	57	15	129	83.82%	75.41%	80.70%	79.17%	0.59
Border (irregular)	57	4	39	29	129	90.70%	66.28%	93.44%	57.35%	0.50
Color (polychromatic)	78	3	27	21	129	90.00%	78.79%	96.30%	56.25%	0.57
Diameter (>6 mm)	81	1	35	12	129	97.22%	87.10%	98.78%	74.47%	0.77
Ugly Duckling Sign	11	4	114	0	129	96.61%	100.00%	73.33%	100.00%	0.83
Little Red Riding Hood Sign	1	0	128	0	129	100.00%	100.00%	100.00%	100.00%	1.00

^a^ Rated positive both by the expert and the non-specialist. ^b^ Rated negative by the expert, positive by the non-specialist. ^c^ Rated negative both by the expert and the non-specialist. ^d^ Rated positive by the expert and negative by the non-specialist.

**Table 2 diagnostics-12-02821-t002:** Results of the dermoscopic assessment by the expert and the non-specialist.

	True Positive ^a^	False Positive ^b^	True Negative ^c^	False Negative ^d^	Total	Specificity	Sensitivity	Positive Predictive Value	Negative Predictive Value	Cohen’s Cappa
Asymmetry	49	11	57	9	126	83.82%	84.48%	81.67%	83.36%	0.68
Atypical network	27	9	76	14	126	89.41%	65.85%	75.00%	84.44%	0.57
Blue veil	8	2	114	2	126	98.28%	80.00%	80.00%	98.28%	0.78

^a^ Rated positive both by the expert and the non-specialist. ^b^ Rated negative by the expert, positive by the non-specialist. ^c^ Rated negative both by the expert and the non-specialist. ^d^ Rated positive by the expert and negative by the non-specialist.

## Data Availability

The data underlying this article are available in the article.

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
