# Peer review of "Melanoma Detection by Non-Specialists: An Untapped Potential for Triage?"

_diagnostics, 2022, doi:10.3390/diagnostics12112821_

Round 1
Reviewer 1 Report
Well written paper, introduction is too long and is missing information about use of dermoscopy among PCPs and APPs in Australia, US. There is no need to describe all imaging modalities and use of AI. It doesn't set the scene for the upcoming study problem.
Discussion ,again, not focusing on the study problem and its outcomes.Authors don't compare their outcomes to other similar studies and don't discuss them. Too much focus on imaging modalities while the goal is not emphasized enough.
Reviewer 2 Report
Honestly, I don’t understand the interest of this paper, We all know that general practitioners use skin imaging techniques in the areas of the globe where skin cancer -especially melanoma - is frequent especially Australia, and USA , it could be interesting maybe in your context, if it is the case, you should state it .
Also, your results are not surprising because the non specialist received a short training in melanoma dermoscopy as you stated in your manuscript , so they will be able to diagnose atypical lesions
Reviewer 3 Report
The study is interesting and well written. It deals with important aspect of public health such as faster and easier detection of melanoma. However it requires some modification
The introduction should me modified. It describes different techniques of artificial intelligence however the aim of the study is comparison of screening efficacy of non- specialist and professional dermatologist
Discussion section: The authors did not mention that this study is the first study. Therefore it would be interesting to compare the obtained results with other literature data.
Conclusions should be rewrite. “ However, combined with the presently evolving artificial intelligence- based diagnostic tools, the inclusion of non-specialists such as general practitioners in melanoma screening might represent an updated potential for triage” –There are no data in the study that support this conclusion
Round 2
Reviewer 2 Report
Dear editor,
I thank you for sending me this manuscript for a second review
the idea of the manuscript is not original as i stated in the first review , but it could be of interest for skin cancer management and triage in Europe